**Data Availability Statement:** All relevant data are within the manuscript and its Supporting Information files.

**Funding:** The author(s) received no specific funding for this work.

# Changes in loneliness prevalence and its associated factors among Bangladeshi older adults during the COVID-19 pandemic

**Sabuj Kanti Mistry** [1,2,3,4] *, **A. R. M. Mehrab Ali** [1], **Uday Narayan Yadav**[2,5,6], **Fouzia Khanam**[7], **Md. Nazmul Huda**[1,8,9]

1 ARCED Foundation, Dhaka, Bangladesh, 2 Centre for Primary Health Care and Equity, University of New South Wales, Sydney, Australia, 3 BRAC James P Grant School of Public Health, BRAC University, Dhaka, Bangladesh, 4 Department of Public Health, Daffodil International University, Dhaka, Bangladesh, 5 National Centre for Epidemiology and Population Health, The Australian National University, Canberra, ACT, Australia, 6 Centre for Research Policy and Implementation, Biratnagar, Nepal, 7 Department of Public Health, North South University, Dhaka, Bangladesh, 8 Translational Health Research Institute, Western Sydney University, Campbeltown, NSW, Australia, 9 The School of Liberal Arts and Social Sciences, Independent University, Dhaka, Bangladesh

* smitra411@gmail.com

## Abstract

### Aims

Worldwide, loneliness is one of the most common psychological phenomena among older adults, adversely affecting their physical and mental health conditions during the COVID-19 pandemic. This study aims to assess changes in the prevalence of loneliness in the two timeframes (first and second waves of COVID-19 in Bangladesh) and identify its correlates in pooled data.

### Methods

This repeated cross-sectional study was conducted on two successive occasions (October 2020 and September 2021), overlapping with the first and second waves of the COVID-19 pandemic in Bangladesh. The survey was conducted remotely through telephone interviews among 2077 (1032 in the 2020-survey and 1045 in the 2021-survey) older Bangladeshi adults aged 60 years and above. Loneliness was measured using the 3-item UCLA Loneliness scale. The binary logistic regression model was used to identify the factors associated with loneliness in pooled data.

### Results

We found a decline in the loneliness prevalence among the participants in two survey rounds (51.5% in 2021 versus 45.7% in 2020; $P$ = 0.008), corresponding to 33% lower odds in the 2021-survey (AOR 0.67, 95% CI 0.54–0.84). Still, nearly half of the participants were found to be lonely in the latest survey. We also found that, compared to their respective counterparts, the odds of loneliness were significantly higher among the participants without a partner (AOR 1.58, 95% CI 1.20–2.08), with a monthly family income less than 5000 BDT (AOR 2.34, 95% CI 1.58–3.47), who lived alone (AOR 2.17, 95% CI 1.34–3.51), with poor

**Competing interests:** The authors have declared that no competing interests exist.

memory or concentration (AOR 1.58, 95% CI 1.23–2.03), and suffering from non-communicable chronic conditions (AOR 1.55, 95% CI 1.23–1.95). Various COVID-19-related characteristics, such as concern about COVID-19 (AOR 1.28, 95% CI 0.94–1.73), overwhelm by COVID-19 (AOR 1.53, 95% CI 1.14–2.06), difficulty earning (AOR 2.00, 95% CI 1.54–2.59), and receiving routine medical care during COVID-19 (AOR 2.08, 95% CI 1.61–2.68), and perception that the participants required additional care during the pandemic (AOR 2.93, 95% CI 2.27–3.79) were also associated with significantly higher odds of loneliness. However, the odds of loneliness were significantly lower among the participants with formal schooling (AOR 0.71, 95% CI 0.57–0.89) and with a family of more than four members (AOR 0.76, 95% CI 0.60–0.96).

## Conclusions

The current study found a decreased prevalence of loneliness among Bangladeshi older adults during the ongoing pandemic. However, the prevalence is still very high. The findings suggest the need for mental health interventions that may include improving social interactions increasing opportunities for meaningful social connections with family and community members and providing psychosocial support to the vulnerable population including older adults during the pandemic. It also suggests that policymakers and public health practitioners should emphasise providing mental health services at the peripheral level where the majority of older adults reside.

## Introduction

Loneliness is an emotional and mental state [1] that an individual faces in terms of subjective feelings of stress [2], sadness, low self-esteem [3], and hopelessness [4]. Loneliness is one of the most common psychological phenomena among older adults worldwide; around one-third of older adults are reported to be lonely [1, 5]. Loneliness increases the risk of heart disease, stroke, mortality, stress, chronic depression, dementia, and suicidal tendencies by damaging physical and mental health [6–9]. Empirical evidence showed that the risk of stroke and dementia raised by 30% and 50%, respectively, due to the sense of loneliness [10]. Loneliness also increased the chance of hospital visits [11], decreased quality of life [12], and mortality [13] among the older population. Additionally, one meta-analysis depicted that loneliness increases the risk of all-cause mortality by 26% among older adults [6]. Thus, loneliness in older adults remains a considerable public health concern.

The long-term COVID-19 pandemic has hard hit the world since early 2020 [14]. Recent studies showed that, amid the contemporary COVID-19 pandemic, a drastic increase in the prevalence of mental health-related issues (such as loneliness, fear, self-harm, frequent mood changes, and suicide) is seen worldwide [15–17]. Moreover, the mortality and associated physical and mental consequences of COVID-19 are disproportionately higher among the older populations [18]. The unpredictable nature of the disease, the fear of getting sick or dying, being stereotyped by others, restricted movements, home confinement for indefinite periods, limited social connectedness, and substantial and growing financial losses could aggravate the psychological situation of the general population, including older adults [19, 20].

The uncertainties and mitigating measures related to the pandemic have changed peoples' everyday lifestyles and social relationships, making them vulnerable to loneliness. The concern about loneliness among older people is particularly worrying due to their living status, need

for long-term care, and weak physical and mental health conditions [21–23]. While social distancing is essential to limit the spread of viral infection, this can negatively affect the sense of social connectedness, ultimately affecting older people's mental health. The consequence could be more severe for those with pre-existing loneliness and mental conditions [24].

Previous studies have documented that sociodemographic characteristics, such as being female, older age and unemployment, are linked to people's loneliness during the pandemic [25–29]. Poor mental and physical health, anxiety and depression can increase loneliness [25, 30, 31]. Besides, chronic conditions and functional disability have been associated with higher perceived loneliness among older adults [25, 26, 28, 29]. One study demonstrated that more worry about COVID-19 infection and more financial strain because of the pandemic was linked to greater loneliness [32]. Conversely, social resources may promote resilience among older adults during the COVID-19 pandemic [33]. Greater household size, having a partner and more emotional support are associated with less loneliness [25–27, 29, 34]. However, there are limited studies that assessed the association of loneliness with socioeconomic (e.g. income, living arrangement, distance to healthcare facility) and COVID-19 related factors, including pre-existing health conditions of COVID-19 patients, feeling concerned about the pandemic, access to medical care and financial challenges during the pandemic among older people in LMICs, including Bangladesh.

This suggests an urgent need to assess the loneliness level among older adults during the COVID-19 pandemic. To fill this need, during the first wave of the pandemic in 2020, we surveyed 1,032 older Bangladeshi adults. In 2021, the second survey was conducted among the same population to observe changes in loneliness prevalence. In the present study, we used these two cross-sectional survey data to assess changes in the prevalence of loneliness across two timeframes (first and second waves of COVID-19 in Bangladesh) and to explore the factors associated with loneliness among the Bangladeshi older population.

## Methods

### Study design and participants

This repeated cross-sectional study was conducted on two successive occasions, i.e., October 2020 and September 2021, overlapping with the first and second waves of the COVID-19 pandemic in Bangladesh. The study was conducted remotely by the Aureolin Research, Consultancy and Expertise Development (ARCED) Foundation (a non-profit organisation based in Bangladesh). The primary challenge for this study was to develop a sampling frame to select participants. Thus, we utilised our pre-established registry, constructed by merging previously completed community-based studies conducted by ARCED Foundationas described in more detail in our previous studies [35–37], which included households from all eight administrative divisions of Bangladesh, as a sampling frame. Considering the 50% prevalence of loneliness with a 5% margin of error at the 95% confidence level, 90% power of the test, and 95% response rate, a sample size of 1096 was calculated. However, during the 2020 survey, 1032 approached eligible participants responded to the study with an overall response rate of approximately 94%. During the 2021 survey, 1045 approached eligible participants responded to the study with an overall response rate of approximately 95%. Based on the population distribution of older adults by geography in Bangladesh, we adopted a probability proportionate to size (of the eight-division) approach to select older adults in each division [38]. The inclusion criterion was the minimum age of 60 years. In each administrative division, households were selected using a simple random sampling technique from the list of eligible participants in the registry. Subsequently, one eligible participant was interviewed from each of the selected households. Hence, the number of included households and respondents is equal. The exclusion criteria included adverse

mental conditions (clinically proven schizophrenia, bipolar mood disorder, dementia/cognitive impairment), a hearing disability, or an inability to communicate.

## Measures

**Outcome measure.** The study's primary outcome was loneliness, measured using the 3-item UCLA Loneliness scale [39]. The three items included: how often do you feel (i) lack of companionship, (ii) left out, and (iii) isolated in the last two weeks. Each item in the scale is measured in terms of 3-item Likert responses: hardly ever (1 point), some of the time (2 points), and often (3 points). The participants were classified as lonely if they answered 'some of the time' or 'often' to any item [1]. Dichotomised loneliness variable is used for all data analyses. We found it a reliable scale, indicated by the high internal consistency (Cronbach's alpha 0.84) among our study participants in the pooled data.

**Explanatory variables.** An extensive literature review guided the selection of explanatory variables [40–45]. Explanatory variables considered in this study were administrative division (Barishal, Chattogram, Dhaka, Mymensingh, Khulna, Rajshahi, Rangpur, Sylhet), age (categorized as 60–69, and ≥70), sex (male/female), marital status (married/without partner), formal schooling (without formal schooling/with formal schooling), family size (≤4 or >4), family monthly income (BDT) (<5,000, 5,000–10,000, >10,000), residence (urban/rural), current occupation (employed/unemployed or retired), living arrangement (living alone or with family), walking distance to the nearest health center (<30 min/≥30 min), memory or concentration probles (no problem/low memory or concentration), suffering from non-communicable chronic conditions (yes/no), feeling concerned about COVID-19 (hardly, sometimes/often), feeling overwhelmed by COVID-19 (hardly, sometimes/often), difficulty in getting food, medicine, and routine medical care during COVID-19 (no/yes), difficulty in earning during COVID-19 (no/yes), perceived that family members are non-responsive (yes/no), and perceived that they required additional care during COVID-19 (yes/no). Self-reported information on non-communicable chronic conditions, such as arthritis, hypertension, heart diseases, stroke, hypercholesterolemia, diabetes, chronic respiratory diseases, chronic kidney disease, and cancer, was collected.

**Data collection tools and techniques.** A pre-tested semi-structured questionnaire was used to collect the information by interviewing the participants remotely using mobile phones. Data collection was accomplished electronically using SurveyCTO mobile app (https://www.surveycto.com/) by trained research assistants, recruited based on previous experience administering health surveys on the electronic platform. The research assistants were trained extensively before the data collection through Zoom meetings.

The English version of the questionnaire was first translated into Bengali language and then back-translated to English by two researchers (SKM, AMA) to ensure the contents' consistency. The questionnaire was then piloted among a small sample (n = 10) of older adults to refine the language in the final version. The tool used in the pilot study did not receive any corrections/suggestions from the participants on the contents that were developed in the local Bengali language.

**Statistical analysis.** The distribution of the variables was assessed through descriptive analyses. Given our variables' categorical nature, Chi-square tests were performed to compare differences in the prevalence of loneliness by explanatory variables, with a 5% level of significance. We used binary logistic regression models to explore the factors associated with dichotomised loneliness in the pooled data. The initial model was run with all potential covariates listed in Table 2. Then, the final model was selected using backward elimination with the Akaike information criterion (AIC). Adjusted odds ratio (aOR) and associated 95% confidence interval (95% CI) are reported in Table 3. We also performed the model diagnostics, such as

multicollinearity, the area under the curve (AUC), and the Hosmer-Lemeshow test in the model. All analyses were performed using the statistical software package Stata (Version 14.0).

### Ethical approval

The study protocol was approved by the Institutional Review Board of the Institute of Health Economics, the University of Dhaka, Bangladesh (Ref: IHE/2020/1037). Verbal informed consent was sought from the participants before administering the survey. Participation was voluntary, and participants did not receive any compensation.

### Patient and public involvement

Patients and/or the public were not involved in developing the research questions, study design, data collection and result dissemination.

## Results

### Characteristics of the participants

Table 1 shows the characteristics of the study participants by survey year. In terms of survey participant coverage, there was a significant difference across geographic areas; for example, the highest coverage was from the Dhaka division in the 2020 Survey, while the highest coverage was from the Khulna division in the 2021 survey. In both surveys, most participants were 60–69 years old, male, married, without formal schooling, unemployed/retired, lived with family, and in rural areas (Table 1). However, participants' characteristics, including sex, marital status, education, and income, were significantly different across the survey years. Compared to the 2020-survey, a considerably lower proportion of participants in the 2021 survey were male (59% vs. 66%), married (77% vs. 81%), and without formal education (52% vs. 58%). The proportion of participants living with family (92% in 2020 vs. 95% in 2021), in rural areas (74% vs. 83%) and proximity to health facility (49% vs. 56%) increased significantly between the survey years. We also noted a significant increase in the reported psychological characteristics, i.e., a higher proportion of participants reported poor memory or concentration, isolation, being overwhelmed with COVID-19, and having difficulty earning and obtaining food during COVID-19 in 2021 compared to 2020 (Table 1).

### Loneliness prevalence in older adults

Table 2 shows the changes in the prevalence of loneliness over time and their variation and association with participants' characteristics. We found a significant decline in the prevalence of loneliness between the two survey years (51.5% versus 45.7%; $P = 0.008$). As seen in Table 2, the proportion of participants experiencing loneliness decreased for almost all the characteristics presented in Table 2.

Overall, the prevalence of loneliness decreased over time among females, among relatively older adults, those having a formal education, participants with higher family income, participants who were unemployed or retired, participants having memory problems, those suffering from non-communicable chronic conditions, participants who were concerned and overwhelmed by COVID-19, who faced difficulty in earning, getting medicine and routine medical care, and who perceived that they need additional care during the pandemic. However, the prevalence of loneliness seems to be static or increased a bit among the participants without a partner and those living alone (Table 2).

**Table 1. Characteristics of the participants (N = 2077).**

| Characteristics | | 2020 survey | | Survey 2021 | | P |
|---|---|---|---|---|---|---|
| | | n | % | n | % | |
| Overall | | 1032 | 100.0 | 1045 | 100.0 | |
| Administrative division | | | | | | |
| | Barishal | 149 | 14.4 | 146 | 14.0 | 0.001 |
| | Chattogram | 137 | 13.3 | 98 | 9.4 | |
| | Dhaka | 210 | 20.4 | 172 | 16.5 | |
| | Mymensingh | 63 | 6.1 | 69 | 6.6 | |
| | Khulna | 158 | 15.3 | 198 | 19.0 | |
| | Rajshahi | 103 | 10.0 | 145 | 13.9 | |
| | Rangpur | 144 | 14.0 | 161 | 15.4 | |
| | Sylhet | 68 | 6.6 | 56 | 5.4 | |
| Age (year) | | | | | | |
| | 60–69 | 803 | 77.8 | 790 | 75.6 | 0.385 |
| | > = 70 | 229 | 22.2 | 255 | 24.4 | |
| Sex | | | | | | |
| | Male | 676 | 65.5 | 620 | 59.3 | 0.004 |
| | Female | 356 | 34.5 | 425 | 40.7 | |
| Marital status | | | | | | |
| | Married | 840 | 81.4 | 799 | 76.5 | 0.006 |
| | Without partner | 192 | 18.6 | 246 | 23.5 | |
| Formal schooling | | | | | | |
| | Without formal schooling | 602 | 58.3 | 540 | 51.7 | 0.002 |
| | With formal schooling | 430 | 41.7 | 505 | 48.3 | |
| Family size | | | | | | |
| | ≤4 | 318 | 30.8 | 347 | 33.2 | 0.243 |
| | >4 | 714 | 69.2 | 698 | 66.8 | |
| Family monthly income (BDT) | | | | | | |
| | <5000 | 145 | 14.1 | 121 | 11.6 | <0.001 |
| | 5000–10000 | 331 | 32.1 | 469 | 44.9 | |
| | >10000 | 556 | 53.9 | 455 | 43.5 | |
| Residence | | | | | | |
| | Urban | 269 | 26.1 | 182 | 17.4 | <0.001 |
| | Rural | 763 | 73.9 | 863 | 82.6 | |
| Current occupation | | | | | | |
| | Employed | 419 | 40.6 | 407 | 39.0 | 0.441 |
| | Unemployed/retired | 613 | 59.4 | 638 | 61.1 | |
| Living arrangement | | | | | | |
| | Living with family | 953 | 92.3 | 992 | 94.9 | 0.016 |
| | Living alone | 79 | 7.7 | 53 | 5.1 | |
| Walking distance to the nearest health centre | | | | | | |
| | <30 minute | 503 | 48.7 | 581 | 55.6 | 0.002 |
| | ≥30 minutes | 529 | 51.3 | 464 | 44.4 | |
| Problem in memory or concentration | | | | | | |
| | No problem | 782 | 75.8 | 676 | 64.7 | <0.001 |
| | Low memory or concentration | 250 | 24.2 | 369 | 35.3 | |
| Suffering from non communicable chronic conditions | | | | | | |
| | No | 424 | 41.1 | 447 | 42.8 | 0.435 |

(*Continued*)

**Table 1.** (Continued)

| Characteristics | | 2020 survey | | Survey 2021 | | P |
|---|---|---|---|---|---|---|
| | | n | % | n | % | |
| | Yes | 608 | 58.9 | 598 | 57.2 | |
| Feeling concerned about COVID-19 | | | | | | |
| | Hardly | 299 | 29.0 | 348 | 33.3 | 0.033 |
| | Sometimes to often | 733 | 71.0 | 697 | 66.7 | |
| Feeling overwhelmed by COVID-19 | | | | | | |
| | Hardly | 370 | 36.4 | 334 | 32.1 | 0.041 |
| | Sometimes to often | 647 | 63.6 | 706 | 67.9 | |
| Difficulty in getting food during COVID-19 | | | | | | |
| | No | 553 | 55.3 | 514 | 49.7 | 0.011 |
| | Yes | 447 | 44.7 | 521 | 50.3 | |
| Difficulty in getting medicine during COVID-19 | | | | | | |
| | No | 733 | 75.3 | 764 | 74.8 | 0.765 |
| | Yes | 240 | 24.7 | 258 | 25.2 | |
| Difficulty in earning during COVID-19 | | | | | | |
| | No | 340 | 37.4 | 274 | 27.7 | <0.001 |
| | Yes | 570 | 62.6 | 714 | 72.3 | |
| Difficulty receiving routine medical care during COVID-19 | | | | | | |
| | No | 644 | 69.6 | 709 | 71.0 | 0.517 |
| | Yes | 281 | 30.4 | 290 | 29.0 | |
| Perceived that family members are non-responsive | | | | | | |
| | No | 687 | 66.6 | 738 | 70.6 | 0.047 |
| | Yes | 345 | 33.4 | 307 | 29.4 | |
| Perceived needing additional care during COVID-19 | | | | | | |
| | No | 769 | 74.5 | 770 | 73.7 | 0.665 |
| | Yes | 263 | 25.5 | 275 | 26.3 | |

1 USD = 85.75 BDT

## Factors associated with loneliness

Table 3 shows the correlates of loneliness in the pooled sample. Compared to the 2020-survey, the odds of loneliness were significantly lower in 2021-survey (AOR 0.67, 95% CI 0.54–0.84). Compared to their respective counterparts, the odds of loneliness were significantly higher among the participants without a partner (AOR 1.58, 95% CI 1.20–2.08), living with a family having a monthly income less than 5000 BDT (AOR 2.34, 95% CI 1.58–3.47), who lived alone (AOR 2.17, 95% CI 1.34–3.51), with poor memory or concentration (AOR 1.58, 95% CI 1.23–2.03), and suffering from non-communicable chronic conditions (AOR 1.55, 95% CI 1.23–1.95). Various COVID-19-related characteristics such as concern about COVID-19 (AOR 1.28, 95% CI 0.94–1.73), overwhelmed by COVID-19 (AOR 1.53, 95% CI 1.14–2.06), difficulty earning (AOR 2.00, 95% CI 1.54–2.59), and receiving routine medical care during COVID-19 (AOR 2.08, 95% CI 1.61–2.68), the perception that they required additional care during the pandemic (AOR 2.93, 95% CI 2.27–3.79) were associated with significantly higher odds of loneliness. However, the odds of loneliness were significantly lower among the participants with formal schooling (AOR 0.71, 95% CI 0.57–0.89) and with a family of more than four members (AOR 0.76, 95% CI 0.60–0.96) (Table 3).

**Table 2. Prevalence of loneliness over time and bivariate analysis (N = 2,077).**

| Characteristics | | 2020 survey | | | 2021 survey | | |
|---|---|---|---|---|---|---|---|
| | | n | %lonely | P[1] | n | %lonely | P[2] |
| Overall | | 1032 | 51.5 | | 1045 | 45.7 | 0.008 |
| Administrative division | | | | | | | |
| | Barishal | 149 | 44.3 | 0.136 | 146 | 43.2 | 0.309 |
| | Chattogram | 137 | 51.1 | | 98 | 43.9 | |
| | Dhaka | 210 | 57.1 | | 172 | 44.2 | |
| | Mymensingh | 63 | 46.0 | | 69 | 39.1 | |
| | Khulna | 158 | 52.5 | | 198 | 46.5 | |
| | Rajshahi | 103 | 49.5 | | 145 | 41.4 | |
| | Rangpur | 144 | 57.6 | | 161 | 53.4 | |
| | Sylhet | 68 | 42.7 | | 56 | 53.6 | |
| Age (year) | | | | | | | |
| | 60–69 | 803 | 49.7 | 0.034 | 790 | 42.8 | 0.001 |
| | > = 70 | 229 | 57.6 | | 255 | 54.5 | |
| Sex | | | | | | | |
| | Male | 676 | 48.5 | 0.009 | 620 | 40.5 | <0.001 |
| | Female | 356 | 57.0 | | 425 | 53.2 | |
| Marital status | | | | | | | |
| | Married | 840 | 50.8 | 0.404 | 799 | 41.2 | <0.001 |
| | Without partner | 192 | 54.2 | | 246 | 60.2 | |
| Formal schooling | | | | | | | |
| | Without formal schooling | 602 | 54.8 | 0.011 | 540 | 51.1 | <0.001 |
| | With formal schooling | 430 | 46.7 | | 505 | 39.8 | |
| Family size | | | | | | | |
| | ≤4 | 318 | 51.9 | 0.853 | 347 | 49.0 | 0.126 |
| | >4 | 714 | 51.3 | | 698 | 44.0 | |
| Family monthly income (BDT) | | | | | | | |
| | <5000 | 145 | 66.2 | <0.001 | 121 | 70.3 | <0.001 |
| | 5000–10000 | 331 | 45.0 | | 469 | 43.7 | |
| | >10000 | 556 | 51.4 | | 455 | 41.1 | |
| Residence | | | | | | | |
| | Urban | 269 | 45.7 | 0.029 | 182 | 42.9 | 0.406 |
| | Rural | 763 | 53.5 | | 863 | 46.2 | |
| Current occupation | | | | | | | |
| | Employed | 419 | 53.7 | 0.233 | 407 | 39.6 | 0.002 |
| | Unemployed/retired | 613 | 49.9 | | 638 | 49.5 | |
| Living arrangement | | | | | | | |
| | Living with family | 953 | 49.7 | <0.001 | 992 | 44.0 | <0.001 |
| | Living alone | 79 | 72.2 | | 53 | 77.4 | |
| Walking distance to the nearest health centre | | | | | | | |
| | <30 minute | 503 | 46.9 | 0.004 | 581 | 46.3 | 0.635 |
| | ≥30 minutes | 529 | 55.8 | | 464 | 44.8 | |
| Problem in memory or concentration | | | | | | | |
| | No problem | 782 | 46.4 | <0.001 | 676 | 38.0 | <0.001 |
| | Low memory or concentration | 250 | 67.2 | | 369 | 59.6 | |
| Suffering from non-communicable chronic conditions | | | | | | | |
| | No | 424 | 40.8 | <0.001 | 447 | 37.1 | <0.001 |

(*Continued*)

**Table 2.** (Continued)

| Characteristics | | 2020 survey | | | 2021 survey | | |
|---|---|---|---|---|---|---|---|
| | | n | %lonely | P[1] | n | %lonely | P[2] |
| | Yes | 608 | 58.9 | | 598 | 52.0 | |
| Feeling concerned about COVID-19 | | | | | | | |
| | Hardly | 299 | 31.4 | <0.001 | 348 | 33.9 | <0.001 |
| | Sometimes to often | 733 | 59.6 | | 697 | 51.5 | |
| Feeling overwhelmed by COVID-19 | | | | | | | |
| | Hardly | 370 | 31.9 | <0.001 | 334 | 35.3 | <0.001 |
| | Sometimes to often | 647 | 62.1 | | 706 | 50.4 | |
| Difficulty in getting food during COVID-19 | | | | | | | |
| | No | 553 | 38.5 | <0.001 | 514 | 36.2 | <0.001 |
| | Yes | 447 | 65.3 | | 521 | 54.5 | |
| Difficulty in getting medicine during COVID-19 | | | | | | | |
| | No | 733 | 44.1 | <0.001 | 764 | 40.8 | <0.001 |
| | Yes | 240 | 70.4 | | 258 | 58.5 | |
| Difficulty in earning during COVID-19 | | | | | | | |
| | No | 340 | 28.8 | <0.001 | 274 | 29.9 | <0.001 |
| | Yes | 570 | 63.9 | | 714 | 50.3 | |
| Difficulty receiving routine medical care during COVID-19 | | | | | | | |
| | No | 644 | 40.1 | <0.001 | 709 | 39.4 | <0.001 |
| | Yes | 281 | 76.2 | | 290 | 60.7 | |
| Perceived that family members are non-responsive | | | | | | | |
| | No | 687 | 52.0 | 0.643 | 738 | 43.1 | 0.010 |
| | Yes | 345 | 50.4 | | 307 | 51.8 | |
| Perceived needing additional care during COVID-19 | | | | | | | |
| | No | 769 | 40.6 | <0.001 | 770 | 40.0 | <0.001 |
| | Yes | 263 | 83.3 | | 275 | 61.5 | |

P[1]: P-value from Pearson Chi-square test comparing participants with and without loneliness in 2020.

P[2]: P-value from Pearson Chi-square test comparing participants with and without loneliness in 2021.

1 USD = 85.75 BDT

## Discussion

This repeated cross-sectional study assessed changes in the loneliness prevalence in older adults and its associated correlates during the first and second waves of the COVID-19 pandemic in Bangladesh. Findings found a decreased loneliness prevalence among older adults during the first and second waves of COVID-19, from 51.5% in 2020 to 45.7% in 2021.

Our study indicates that loneliness in older people during COVID-19 is still very high. To compare our study's findings, we did not find any published study that reported changes in older people's loneliness during the COVID-19 pandemic in Bangladesh and other low- and middle-income countries. However, loneliness prevalence among older adults (45.70%) in our study is broadly comparable with the rates reported by studies from high-income countries, including the United States (54%) [46], the United Kingdom (35.86%) [47], and Canada (43.1%) [18]. Meanwhile, a cross-sectional study that was conducted online on Columbian women (aged 40–59 years) found a similar rate of loneliness (44.5%) during the COVID-19 pandemic to that of the current study [48]. Pre-pandemic evidence found prevalence rates in older adults (aged ≥60 years) in Bangladesh [49] and Singapore [50], ranging from 42% to 54.3%. Many factors, including study design, study tools to measure loneliness, different time

**Table 3. Factors associated with loneliness among the participants (N = 2077).**

| Characteristics | | aOR | 95% CI | P |
|---|---|---|---|---|
| Survey year | | | | |
| | 2020 | Ref | | |
| | 2021 | 0.67 | 0.54–0.84 | 0.001 |
| Age (year) | | | | |
| | 60–69 | Ref | | |
| | > = 70 | 1.18 | 0.90–1.54 | 0.237 |
| Marital status | | | | |
| | Married | Ref | | |
| | Without partner | 1.58 | 1.20–2.08 | 0.001 |
| Formal schooling | | | | |
| | Without formal schooling | Ref | | |
| | With formal schooling | 0.71 | 0.57–0.89 | 0.003 |
| Family size | | | | |
| | ≤4 | Ref | | |
| | >4 | 0.76 | 0.60–0.96 | 0.021 |
| Family monthly income (BDT) | | | | |
| | >10000 | Ref | | |
| | 5000–10000 | 1.01 | 0.80–1.29 | 0.907 |
| | <5000 | 2.34 | 1.58–3.47 | <0.001 |
| Living arrangement | | | | |
| | Living with family | Ref | | |
| | Living alone | 2.17 | 1.34–3.51 | 0.002 |
| Walking distance to the nearest health centre | | | | |
| | <30 minute | Ref | | |
| | ≥30 minutes | 1.18 | 0.95–1.47 | 0.130 |
| Problem in memory or concentration | | | | |
| | No problem | Ref | | |
| | Low memory or concentration | 1.58 | 1.23–2.03 | <0.001 |
| Suffering from non-communicable chronic conditions | | | | |
| | No | Ref | | |
| | Yes | 1.55 | 1.23–1.95 | <0.001 |
| Feeling concerned about COVID-19 | | | | |
| | Hardly | Ref | | |
| | Sometimes to often | 1.28 | 0.94–1.73 | 0.113 |
| Feeling overwhelmed by COVID-19 | | | | |
| | Hardly | Ref | | |
| | Sometimes to often | 1.53 | 1.14–2.06 | 0.005 |
| Difficulty in earning during COVID-19 | | | | |
| | No | Ref | | |
| | Yes | 2.00 | 1.54–2.59 | <0.001 |
| Difficulty receiving routine medical care during COVID-19 | | | | |
| | No | Ref | | |
| | Yes | 2.08 | 1.61–2.68 | <0.001 |
| Perceived that family members are non-responsive | | | | |
| | No | Ref | | |
| | Yes | 1.25 | 0.98–1.59 | 0.068 |
| Perceived needing additional care during COVID-19 | | | | |

*(Continued)*

**Table 3.** (Continued)

| Characteristics | | aOR | 95% CI | P |
|---|---|---|---|---|
| | No | Ref | | |
| | Yes | 2.93 | 2.27–3.79 | <0.001 |

1 USD = 85.75 BDT

frames, pre-and post-pandemic factors, sampling differences, cultural factors, and socio-economic contexts, may elucidate the differences in older people's loneliness during the pandemic.

The present study reported that loneliness was higher among older adults with a family with limited income, difficulty in earning and living alone. The declining traditional value system wherein family members live together and care for each other [51] and the increasing trend of nuclear families [52] are the potential reasons for older people's loneliness in Bangladesh. Our study's findings are reasonably comparable with existing literature, suggesting increased loneliness among older people with children, limited income and living alone in the community [53]. We also found a higher prevalence of loneliness in older people with no partners, those having poor memory or concentration and those suffering from non-communicable chronic conditions. Older people's loneliness was expected because they tend to live alone, lose spouses, family and friends [46, 54], and face chronic illness and experience hearing loss more than younger individuals [55]. The current study's increased loneliness in older adults may be due to COVID-19-related physical distancing and isolation measures [53]. Furthermore, limited participation in religious activities [56], limited regular activities (e.g. physical activity, exercise etc.) [57, 58], and quality of interpersonal relationships with them [48] might explain such increased loneliness in older adults. Our findings suggest loneliness prevention intervention that can address specific needs of older adults such as health conditions, housing environment, level of connectedness to close ones and cultural characteristics, the degree of loneliness experienced and the available supportive environment required for maintaining a good quality of life.

Our study found that participants feeling concerned about and overwhelmed by COVID-19, those perceiving that they required additional care, and receiving routine medical care during the pandemic were more likely to feel lonely. We could not find any related studies that had been conducted on loneliness in older adults in Bangladesh and beyond. Thus, to our knowledge, there are no or limited data available for comparison in this context. However, evidence indicates that the emergency of coronavirus interrupted [59] and overwhelmed individuals' lives, thus requiring additional care during the pandemic [44, 60]. Furthermore, people's access to public transport was limited due to coronavirus-related lockdowns [59]. As most people in Bangladesh primarily depend on public transport, such limited transportation services interrupted their routine medical care [61]. Lockdowns during the pandemic may also increase individuals' sedentary behaviours and limit physical activity [62], heightening their tension, anxiety, and fear [48] and resulting in adverse psychological health [63]. During the pandemic, such changes in individuals' everyday lives may increase the risk of loneliness in older adults [5]. Our findings suggest providing targeted care and services for reducing loneliness in older adults while maintaining Covid-1-related physical distancing measures.

To our knowledge, using a repeated cross-sectional survey, this is the first study examining the association of loneliness in older adults with their schooling and family size during the COVID-19 pandemic. As expected, decreased loneliness was found among participants with formal schooling and a family of more than four members in our study and previous research

also documented similar results [33–35]. The potential reasons are that participants with formal schooling are educated and may be aware of the adverse effects of loneliness, and the potential strategies for preventing and responding to loneliness [64]. Education can minimise individuals' loneliness by enhancing social networks and connectedness with friends and external individuals via social media [65]. Furthermore, older people with more family members have the potential to mix and interact with many family members, which reduces their risk of being lonely.

## Strengths and limitations of the study

The current study has several strengths. First, this study is among the first in Bangladesh to estimate the changes in loneliness prevalence and examine its correlates among Bangladeshi older adults during the two timeframes (first and second waves of COVID-19 in Bangladesh). Our study contributed to the limited international literature [18, 46, 47, 66] that has examined loneliness prevalence among older adults and its predictors during the COVID-19 pandemic. Second, to the best of our knowledge, some of the correlates of loneliness in older adults (e.g. feeling overwhelmed by the COVID-19 pandemic, perceiving that they required additional care during the pandemic, and receiving routine medical care) in the current study have been reported for the first time in Bangladesh and globally. Despite these strengths, our study's findings have several limitations. First, our research was cross-sectional in nature. Therefore, causality cannot be established. Second, our study is limited to quantitative analysis, as we did not explore the qualitative aspects of older adults' feelings of loneliness during the first and second waves of the pandemic. Secondly, amidst the pandemic, the sampling frame for the study was constructed by merging the contact information of previously completed community-based studies conducted by ARCED Foundation. Also, we had to conduct telephone interviews, and the sample might likely exclude those who do not have telephone access. Therefore, our sample may not represent the entire older adults of Bangladesh. These limitations highlight the need for future nationally representative research with a mixed-method approach, including a qualitative study exploring older adults' experience of loneliness and its associated factors during the COVID-19 pandemic. This will provide a better understanding of older adults' feelings of loneliness and the related factors during the COVID-19 pandemic in Bangladesh.

## Implications for policy and practice

Our findings have several implications for future research and policy for older people in Bangladesh. Firstly, our study's findings highlight the necessity for interventions to engage older adults in social and community activities and improve social interactions and community participation (while practising safety measures to limit the COVID-19 spread) to minimise the likelihood of experiencing loneliness. Secondly, the current study emphasises that it is vital to reduce the higher loneliness prevalence among older people with limited income, difficulty earning and living alone by providing financial support. Older people's loneliness can also be reduced by encouraging their children to contact and look after their parents and provide them with food and shelter. In this context, implementing the existing Parents Care Act (2013) [67] may help older people, including those who are poor, live alone, have earning difficulty, and integrate into their families. Such integration may help older people interact and live with their family members, including children and grandchildren, thus reducing their loneliness. Thirdly, it is equally important to change lonely older people's perceptions and feelings of disconnection during the pandemic. Disseminating loneliness prevention strategies via information sessions and leaflets might be a potential option for minimising older people's loneliness during the pandemic.

## Conclusion

The present study revealed that while the prevalence of loneliness decreased among Bangladeshi older adults during the pandemic, still nearly half of the participants were lonely, which needs to be taken seriously. The study's findings suggest the need for designing and implementing people-centered supportive mental health interventions for older adults to improve social interactions increasing opportunities for meaningful social connections with family and community members during this pandemic and beyond. While designing interventions, addressing the factors associated with loneliness is crucial. Policymakers and health care practitioners should also consider strengthening social support care and providing psychosocial support for the older population as part of the emergency management plan during this COVID-19 pandemic.

## Supporting information

**S1 Data.**
(DTA)

## Acknowledgments

We acknowledge the role of Sadia Sumaia Chowdhury, Programme Manager, ARCED Foundation and Md. Zahirul Islam, Project Associate, ARCED Foundation, for their support in data collection for the study.

## Author Contributions

**Conceptualization:** Sabuj Kanti Mistry, A. R. M. Mehrab Ali, Uday Narayan Yadav.

**Data curation:** Sabuj Kanti Mistry.

**Formal analysis:** Sabuj Kanti Mistry.

**Investigation:** Sabuj Kanti Mistry.

**Methodology:** Sabuj Kanti Mistry.

**Project administration:** Sabuj Kanti Mistry, A. R. M. Mehrab Ali.

**Supervision:** Sabuj Kanti Mistry, A. R. M. Mehrab Ali.

**Validation:** Sabuj Kanti Mistry.

**Writing – original draft:** Sabuj Kanti Mistry, A. R. M. Mehrab Ali, Uday Narayan Yadav, Fouzia Khanam, Md. Nazmul Huda.

**Writing – review & editing:** Sabuj Kanti Mistry, Uday Narayan Yadav, Md. Nazmul Huda.

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
