## [Decision Letter · Decision Letter 0]

31 Aug 2022

PONE-D-22-03258Loneliness is decreasing in Bangladeshi older adults during the COVID-19 pandemic, still the prevalence is high: A repeated cross-sectional study’s findingsPLOS ONE

Dear Dr. Mistry,

Thank you for submitting your manuscript to PLOS ONE. After careful consideration, we feel that it has merit but does not fully meet PLOS ONE’s publication criteria as it currently stands. Therefore, we invite you to submit a revised version of the manuscript that addresses the points raised during the review process.

The manuscript is interesting specially capturing adult loneliness during pandemic period. However, the methodology and analytical parts need to revised or re-assessed before being consider for further thoughts. Furthermore, the title and sample strategy need to be reconsidered.  

We look forward to receiving your revised manuscript.

Kind regards,

Mahfuzar Rahman, MD, PhD

Academic Editor

PLOS ONE

Journal Requirements:

Reviewers' comments:

Reviewer's Responses to Questions

**Comments to the Author**

1. Is the manuscript technically sound, and do the data support the conclusions?

Reviewer #1: Yes

Reviewer #2: Partly

2. Has the statistical analysis been performed appropriately and rigorously? 

Reviewer #1: No

Reviewer #2: Yes

3. Have the authors made all data underlying the findings in their manuscript fully available?

Reviewer #1: Yes

Reviewer #2: No

4. Is the manuscript presented in an intelligible fashion and written in standard English?

Reviewer #1: Yes

Reviewer #2: Yes

5. Review Comments to the Author

Reviewer #1: Thanks to the authors for addressing an important public health issue in the context of Bangladesh. Bangladesh is a poor developing country where this issue is often neglected but in developed countries it gets much importance. I am satisfied with the version, clear and concise. I personally know this team who are relentlessly working in different public health issues in Bangladesh that was unexplored earlier.

However, I have some comments regarding the paper which are mostly addressable.

What is the power of the study?

Why authors chose binary logistics regression since the scale gives you quantitative value, it would fit linear regression much better than only binary logistics regression?

The conclusion of the abstract brought some generic comments do you have any specific suggestions how to reduce loneliness among older people in Bangladesh.

Most of the discussion written comparing to USA Canada UK they’re the highest developed countries, please establish the arguments with some Asian countries.

Why authors doesn’t apply multilevel analysis since the data was collected from different heterogeneous divisions? It will suit better for reducing cluster effects.

The title looks odd, please rework on the title.

Reviewer #2: This is a fairly standard analysis of two separate cross-sectional telephone surveys of loneliness among adults 60+ in Bangladesh that overlapped with the first and second wave or the COVID -19 in Bangladesh. Subjects were recruited from a pre-established household registry developed by the Aureolin Research, Consultancy and Expertise Development Foundation (ARCED). To be eligible to participate in the survey participants had to be a minimum of 60 years of age and free of common mental and cognitive disorders, not have a hearing disability or an inability to communicate. The study found the level of loneliness decrease between Time 1 and Time 2. The odds of loneliness were significantly higher people among who were without a partner, had a low income a who lived alone, had poor memory or concentration and suffered from non-communicable chronic disease. The odds of loneliness were significantly lower among individuals with formal education

While the manuscript and the findings interesting there are some concerns that the authors need to respond to:

1) How representative are the samples of the Bangladeshi population?

2) Are there censuses which could be used to compare the representativeness of the survey samples?

3) Where does the population registry that the survey samples are drawn come from?

4) What is ARCED? Profit/non-profit?

5) How specifically were the survey samples recruited from the existing registry?

6) Are participants paid?

7) Since this is a telephone survey, what is the rate of telephone ownership in Bangladesh? Where cell/mobile phones or land lines used?

8) Are the differences found in loneliness found between the initial and second cross-sectional surveys a result of difference in the composition of the samples at T1 and T2?

Minor Points:

Page 6 1 st paragraph, line 2 - the UCLA Loneliness scale - as indicated by the rest of the paragraph this scale does not have Yes/No responses but rather has set of Likert types response.

Page 6 2nd paragraph - line 1 - "literatureguided" - the words need to be separated.

" line 3 "rangpur' Needs to be capitalized

Page 15 last paragraph line 3 "anyrelated" the words need to be separated.

6. PLOS authors have the option to publish the peer review history of their article (what does this mean?). If published, this will include your full peer review and any attached files.

Reviewer #1: No

Reviewer #2: No

---

## [Author Response · Author response to Decision Letter 0]

7 Sep 2022

Please see the attachment "Review response_R1"

---

## [Editor Report · Decision Letter 1]

24 Oct 2022

Changes in loneliness prevalence and its associated factors among Bangladeshi older adults during the COVID-19 pandemic

PONE-D-22-03258R1

Dear Dr. Mistry,

We’re pleased to inform you that your manuscript has been judged scientifically suitable for publication and will be formally accepted for publication once it meets all outstanding technical requirements.

Kind regards,

Mahfuzar Rahman, MD, PhD

Academic Editor

PLOS ONE